# RHBDL4-triggered downregulation of COPII adaptor protein TMED7 suppresses TLR4-mediated inflammatory signaling

Julia D. Knopf[1,2,9], Susanne S. Steigleder [1,2,9], Friederike Korn [1,2], Nathalie Kühnle[1], Marina Badenes [3,8], Marina Tauber[2], Sebastian J. Theobald[4,5,6], Jan Rybniker[4,5,6], Colin Adrain [3,7] & Marius K. Lemberg [1,2] ✉

The toll-like receptor 4 (TLR4) is a central regulator of innate immunity that primarily recognizes bacterial lipopolysaccharide cell wall constituents to trigger cytokine secretion. We identify the intramembrane protease RHBDL4 as a negative regulator of TLR4 signaling. We show that RHBDL4 triggers degradation of TLR4's trafficking factor TMED7. This counteracts TLR4 transport to the cell surface. Notably, TLR4 activation mediates transcriptional upregulation of RHBDL4 thereby inducing a negative feedback loop to reduce TLR4 trafficking to the plasma membrane. This secretory cargo tuning mechanism prevents the over-activation of TLR4-dependent signaling in an in vitro *Mycobacterium tuberculosis* macrophage infection model and consequently alleviates septic shock in a mouse model. A hypomorphic RHBDL4 mutation linked to Kawasaki syndrome, an ill-defined inflammatory disorder in children, further supports the pathophysiological relevance of our findings. In this work, we identify an RHBDL4-mediated axis that acts as a rheostat to prevent over-activation of the TLR4 pathway.

Protein secretion is an essential process that delivers critical components to the plasma membrane and the extracellular milieu. Cells rapidly and precisely regulate the composition and flux of secretory vesicles, while deregulated protein secretion contributes to various human diseases[1]. One of the key functions of the secretory pathway is to control signaling by regulating protein abundance. This is primarily achieved by the ratio of synthesis at the endoplasmic reticulum (ER) to degradation, either via the lysosomal or the proteasomal degradation route[2]. The latter relies on the ER-associated degradation (ERAD) pathway, which, assisted by several E3 ubiquitin ligases

and the AAA + -ATPase p97, targets proteins from the ER towards the proteasome[3,4]. Thereby ERAD serves as the primary clearance mechanism for misfolded or 'orphan' proteins but also plays a crucial role in regulatory abundance control. While transcriptional changes typically occur in the range of hours, in many cases, dynamic adaptation of the secretion rate is faster by several orders of magnitude, underscoring the importance of other regulatory measures. Our previous work in yeast and mammalian cells points towards a key role of ERAD-linked proteases in the control of secretion dynamics[5,6]. RHBDL4 is such an ER-resident intramembrane protease, which by a

[1]Center for Molecular Biology of Heidelberg University (ZMBH), Heidelberg, Germany. [2]Center for Biochemistry and Cologne Excellence Cluster on Cellular Stress Responses in Aging-Associated Diseases (CECAD), Faculty of Medicine, University of Cologne, Cologne, Germany. [3]Instituto Gulbenkian de Ciência (IGC), Oeiras, Portugal. [4]Department I of Internal Medicine, Faculty of Medicine and University Hospital Cologne, University of Cologne, 50937 Cologne, Germany. [5]Center for Molecular Medicine Cologne (CMMC), University of Cologne, 50931 Cologne, Germany. [6]German Center for Infection Research (DZIF), Partner Site Bonn-Cologne, 50931 Cologne, Germany. [7]Patrick G Johnston Centre for Cancer Research, Queen's University Belfast, Belfast, UK. [8]Present address: Faculty of Veterinary Medicine, Lusofona University and Faculty of Veterinary Nursing, Polytechnic Institute of Lusofonia, Lisbon, Portugal. [9]These authors contributed equally: Julia D. Knopf, Susanne S. Steigleder. ✉e-mail: m.lemberg@uni-koeln.de

so far unknown mechanism, impacts ER-export and non-canonical secretion of several membrane-anchored bioactive molecules, including the pro-form of the transforming growth factor α (proTGFα) and CD44[6].

RHBDL4 is one of four mammalian rhomboid proteases in the secretory pathway[7], which in addition to the conserved rhomboid fold harboring the catalytically active serine-histidine dyad, has a conserved ubiquitin interacting motif (UIM) and a valosin-containing protein (VCP)/p97-binding motif (VBM) in its C-terminal domain[8]. Cell-based assays probing a set of model substrates previously revealed that RHBDL4 cleaves unstable proteins either in their transmembrane (TM) domain or in their luminal portion[8]. As a consequence of cleavage by RHBDL4, protein fragments are extracted in a p97-dependent manner by the ERAD machinery[8]. In addition to membrane proteins, RHBDL4 also cleaves aggregation-prone proteins devoid of any membrane anchor[9], and, by ill-defined mechanisms, induces ER-export and protein secretion[6,10,11]. RHBDL4 activity is regulated both at transcriptional and post-translational levels. As part of the unfolded protein response (UPR), RHBDL4 is transcriptionally upregulated[8]. Besides, two structural features within RHBDL4: the protease's cytoplasmic ubiquitin-interacting motif (UIM)[8], and two putative membrane-embedded cholesterol binding sites[12] have been described to control substrate turnover. In addition, the RHBDL4-induced non-canonical secretion of proTGFα is modulated by G-protein coupled receptors and protein kinase C[6]. It was shown that RHBDL4 is phosphorylated at eight sites at its cytoplasmic C-terminus[13], suggesting that major signaling pathways may modulate RHBDL4 activity. Taken together, while the past decade has revealed several substrates and modes of regulation for RHBDL4, much remains to be learned about its complete substrate range, physiological regulation, and function.

The relative distribution of proteins along the secretory pathway is mainly the product of selective cargo recognition and sorting. In the ER, cargo selection and export depend on selective recruitment into and subsequent anterograde trafficking by COPII (for 'coat protein complex II')-coated vesicles[14–16]. Cargo loading into COPII vesicles is tightly regulated. Some membrane proteins have defined cytosolic sorting signals that directly bind to the COPII coat subunits[17,18]. However, many proteins are devoid of such sorting signals; hence the efficient sorting of most proteins into COPII vesicles relies on cargo receptors, including the p24 protein family (for review, see[19]). Humans express nine distinct p24 proteins (also known as TMED, for 'transmembrane emp24 protein transport domain-containing), which can assemble into different homo- and hetero-oligomers and serve as cargo receptors for a diverse set of proteins[20,21].

TMED7, a key member of the p24 family, is a cargo receptor for the toll-like receptor 4 (TLR4)[22], the central pattern-recognition receptor that coordinates the inflammatory response to the bacterial cell wall constituent lipopolysaccharide (LPS)[23]. Since TLR4 activation elicits strong inflammatory responses and is associated with pathological conditions in infectious diseases, in-particular sepsis[24], the threshold and duration of TLR4 signaling must be tightly regulated. Consequently, a wide range of mechanisms affect TLR4 expression, subcellular transport, activation, and eventually degradation[25]. As noted above, TMED7, which directly interacts with newly synthesized TLR4 in the ER, serves as a TLR4 cargo receptor. Overexpression of TMED7 can boost TLR4 signaling, while its silencing has the opposite effect. This suggests that modulation of TMED7 levels may play a critical role in the regulation of TLR4 signaling by controlling the loading of TLR4 into COPII vesicles and its subsequent anterograde trafficking[22].

Here, we identify by a proteomic screen TMED7 as endogenous RHBDL4 substrate, whose proteolytic turnover by RHBDL4 dampens TLR4 signaling under inflammatory conditions. Our work identifies a secretory cargo abundance control mechanism mediated by intramembrane proteolysis, exerting control on a critical signaling pathway.

## Results

### RHBDL4 cleaves certain p24 proteins to trigger their degradation

Recent evidence indicates that RHBDL4 protease activity plays a role in tuning secretion dynamics in tissue culture cells[6]. However, it remained unclear which substrate is linked to this phenomenon and what the underpinning mechanism was. Although we recently applied a substrate trapping approach to reveal that RHBDL4 cleaves endogenous oligosaccharyltransferase subunits to modulate glycosylation activity[26], no other more overt molecular link to protein trafficking has been observed so far. To further investigate the native substrate spectrum of RHBDL4, we adopted a distinct approach, analyzing the membrane proteome of HEK293T cells lacking RHBDL4 using stable-isotope labeling by amino acids in cell culture (SILAC) and quantitative mass spectrometry (Fig. 1a). As our previous analysis showed that for processing of endogenous substrates, modest changes of the steady-state level can have clear phenotypic consequences[26], we investigated ER-localized proteins that showed even a mild enrichment in knockout cells by candidate testing (Supplementary Data 1). Interestingly, among the membrane proteins that showed such a modest increase in their steady-state level were several TMED/p24 proteins (Supplementary Data 1). Western blot validation of the endogenous protein levels of selected TMED/p24 proteins in total cell lysates of HEK293T RHBDL4 knockout cells compared to wild-type (wt) cells confirmed this finding (Fig. 1b). While the effect for TMED10 did not reach significance, TMED2 and TMED7 showed a clear enrichment in RHBDL4 knockout cells compared to wt, of 46% and 73%, respectively (Fig. 1b). Notably, the mRNA levels of TMED2 and TMED7 were not transcriptionally upregulated, bur a modest downregulation was observed (Supplementary Fig. 1A). Together with the observed steady-state level increase, these results indicate that TMED2, TMED7, and TMED10 are subjects of an RHBDL4-dependent abundance control mechanism.

To test whether RHBDL4 directly cleaves p24 proteins, we set up a cell-based rhomboid assay that has been established to discriminate substrates from non-substrates[27]. To allow sensitive detection of cleavage fragments, we expressed all nine p24 family members as triple FLAG-tagged TMED/p24 constructs alone or with RHBDL4. As RHBDL4-mediated proteolysis is typically coupled to ERAD[8], cells were treated with the proteasome inhibitor MG132 to block potential turnover of cleavage products. Of the nine human TMED/p24 proteins, most were cleaved by ectopically expressed RHBDL4 with the following order of cleavage efficiency TMED7 > TMED3 > TMED6 > TMED2 > TMED1 > TMED10 > TMED5 > TMED4 (Fig. 1c and Supplementary Fig. 1B). While quantification showed a clear change of the full-length to cleavage product ratio for TMED7, TMED3, and TMED6, processing of the other p24 proteins appears to occur to a minor extent only (Fig. 1c). Furthermore, no cleavage fragment was detected for TMED9 (Supplementary Fig. 1B), indicating that only a subset of p24 proteins are physiological RHBDL4-substrates. Consistent with a direct effect on TMED7 mediated by intramembrane proteolysis, co-expression of RHBDL4 wt but not of the S144A active site mutant (SA) generated an N-terminal TMED7 cleavage fragment (Fig. 1d) In addition, increased steady-state level of TMED7 in RHBDL4 knockout cells was rescued by re-expression of the active protease but not by the SA mutant (Fig. 1e). Comparison of the RHBDL4-generated cleavage fragment with truncated TMED7 constructs revealed that RHBDL4-catalyzed cleavage occurs close to serine-149, several amino acids away from the TM anchor within the luminal domain of TMED7 (Supplementary Fig. 1C). Hence, RHBDL4-catalyzed cleavage separates the luminal substrate binding domain of TMED7 from the cytosolic COPII-recruitment motif rendering the resulting fragments unfunctional. A related RHBDL4-mediated clipping mechanism that targets luminal domains for ERAD had been observed for other RHBDL4 substrates before[8]. Consistent with this, the steady-state level of the 22-kDa TMED7 cleavage fragment was

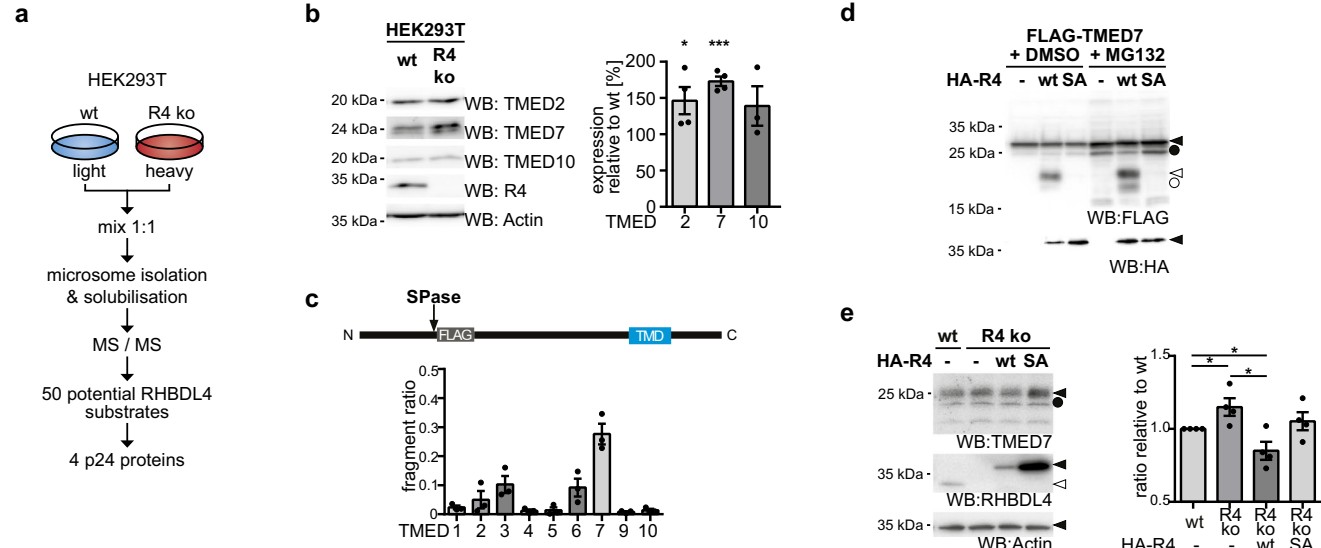

**Fig. 1 | RHBDL4 cleaves certain TMED/p24 proteins to trigger their degradation along the ERAD pathway. a** Experimental outline of the SILAC-based organelle proteomics using HEK293T wt and RHBDL4 knockout cells (R4 ko). **b** Western blot (WB) analysis of endogenous TMED2 ($n = 4$, $p = 0.048$), TMED7 ($n = 4$, $p = <0.0001$), and TMED10 ($n = 3$) in total cell lysates of HEK293T wt or R4 ko cells. Actin is used as a loading control for the TMED2 WB; signals for the other antibodies were obtained from separate WBs. Right panel, WB quantification (means ± SEM; *$p < 0.05$, ***$p < 0.001$, two-sided Student's $t$ test). **c** HEK293T cells were co-transfected with the indicated FLAG-tagged TMED7/p24 constructs and either vector control or HA-tagged RHBDL4. Relative cleavage efficiency was determined. See Supplementary Fig. 1B for representative WB (means ± SEM, $n = 3$). Above: Outline of FLAG-TMED constructs. SPase, site of predicted signal peptide cleavage; TMD, transmembrane domain. **d** RHBDL4 wt but not the SA active site mutant generates an N-terminal FLAG-tagged TMED7 cleavage fragment (open arrow) that is degraded by the proteasome, as shown by an increased steady-state level upon MG132 (2 μM) treatment compared to vehicle control (DMSO). Inhibition of the proteasome also stabilized deglycosylated full-length TMED7 (filled circle) and rhomboid-induced cleavage fragment (open circles) ($n = 3$). **e** Endogenous TMED7 level in HEK293 T-REx R4 ko cells increases when compared to HEK293 T-REx wt cells. Stably expressed HA-R4 wt but not SA active site mutant rescues the increase. Actin is used as a loading control. Lower panel, WB quantification (means ± SEM, $n = 4$; *$p < 0.05$, two-sided Student's $t$ test, wt vs. R4ko $p = 0.0469$, R4ko vs. R4ko wt $p = 0.0129$, wt vs. R4ko wt $p = 0.0498$). Source data are provided as a Source Data file.

sensitive to MG132 (Fig. 1d) and to the p97 inhibitor CB-5083 (Supplementary Fig. 1D). These results indicate that an RHBDL4-dependent ERAD mechanism regulates the abundance of TMED7.

**RHBDL4 specifically interacts with TMED7 in the luminal domain**
To further study how RHBDL4 recognizes TMED7 as a substrate, we extended our cell-based analysis. Previously, we showed that the catalytically inactive RHBDL4 SA mutant can bind to, but not cleave cognate substrates and that this interaction is reduced after mutating the conserved UIM in the C-terminal cytoplasmic domain of RHBDL4[8,26]. Therefore, we used this trapping strategy as an additional readout to characterize how RHBDL4 interacts with TMED7. Hence, we co-expressed FLAG-tagged TMED7 together with GFP-tagged RHBDL4 wt, SA, or the RHBDL4-SA-UIM double mutant and performed affinity purification of the GFP-fusion proteins from detergent-solubilized cells. As observed before, co-expression of RHBDL4 wt resulted in the generation of a cleavage fragment, however no interaction between RHBDL4 wt and full-length TMED7 was detected (Fig. 2a). In contrast, TMED7 was co-purified with the SA and, to a reduced extent, with the SA-UIM double mutant. Interestingly, several higher molecular weight forms appeared after immunoprecipitation with the SA mutant, which were also visible at a reduced level in the input. As they are strongly reduced in the immunoprecipitation with the SA-UIM mutant as bait, they might correspond to ubiquitinated TMED7. To test for ubiquitination of TMED7, we affinity-purified TMED7 from HEK293T cells transfected with either mock plasmid or the SA mutant (Supplementary Fig. 2). Western blot analysis probing with an antibody against ubiquitin revealed several higher molecular weight bands corresponding to ubiquitinated TMED7, which were enriched in the presence of the trapping competent SA mutant. We previously showed that the E3 ubiquitin ligase gp78 acts in concert with RHBDL4 for substrate recruitment and turnover[8]. In order to investigate a potential

contribution of gp78 to TMED7 ubiquitination and RHBDL4-dependent abundance control, we performed substrate trapping upon knockdown of gp78 (Fig. 2b). This revealed a significant reduction of ubiquitinated TMED7 forms trapped by the RHBDL4 SA mutant. In light of the efficient knockdown of gp78, this partial reduction hints at a redundant role for other E3 ubiquitin ligases in TMED7 ubiquitination, which remain to be discovered.

To narrow down features that render TMED7 as an RHBDL4 substrate, we aimed to compare TMED7 to the non-substrate TMED9 (Supplementary Fig. 1B). To this end, we generated chimeras with swapped luminal-, TM-, and cytoplasmic domains of TMED7 and TMED9 (Fig. 2c, d). Indeed, all constructs harboring the TMED7 luminal domain were efficiently cleaved by ectopically expressed RHBDL4, whereas constructs with the TMED9 ectodomain resisted processing. This was independent of the origin of the TM and cytoplasmic domains, indicating that for p24 proteins, the luminal domain discriminates RHBDL4 substrates from non-substrate homologs. Consistent with this swapping, a region of 20 amino acids surrounding the TMED7 cleavage site was sufficient for processing by RHBDL4 when inserted into TMED9. By contrast, cleavage was abrogated when the respective amino acids were replaced in TMED7 with the equivalent stretch in TMED9 (Fig. 2e). Likewise, replacing TMED7's scissile peptide bond residues serine-149 and alanine-150 with proline significantly reduced the processing of the substrate (Fig. 2f). These results reveal that specific features in the luminal domain of TMED7 determine its processing and degradation by an RHBDL4-dependent ERAD pathway.

**RHBDL4 ablation stabilizes TMED7 and increases TLR4 signaling**
Next, we asked whether the RHBDL4-mediated abundance changes affect TMED7 dependent secretion dynamics. As TMED7 has been shown to play a specific role in cell surface trafficking of TLR4[22], we

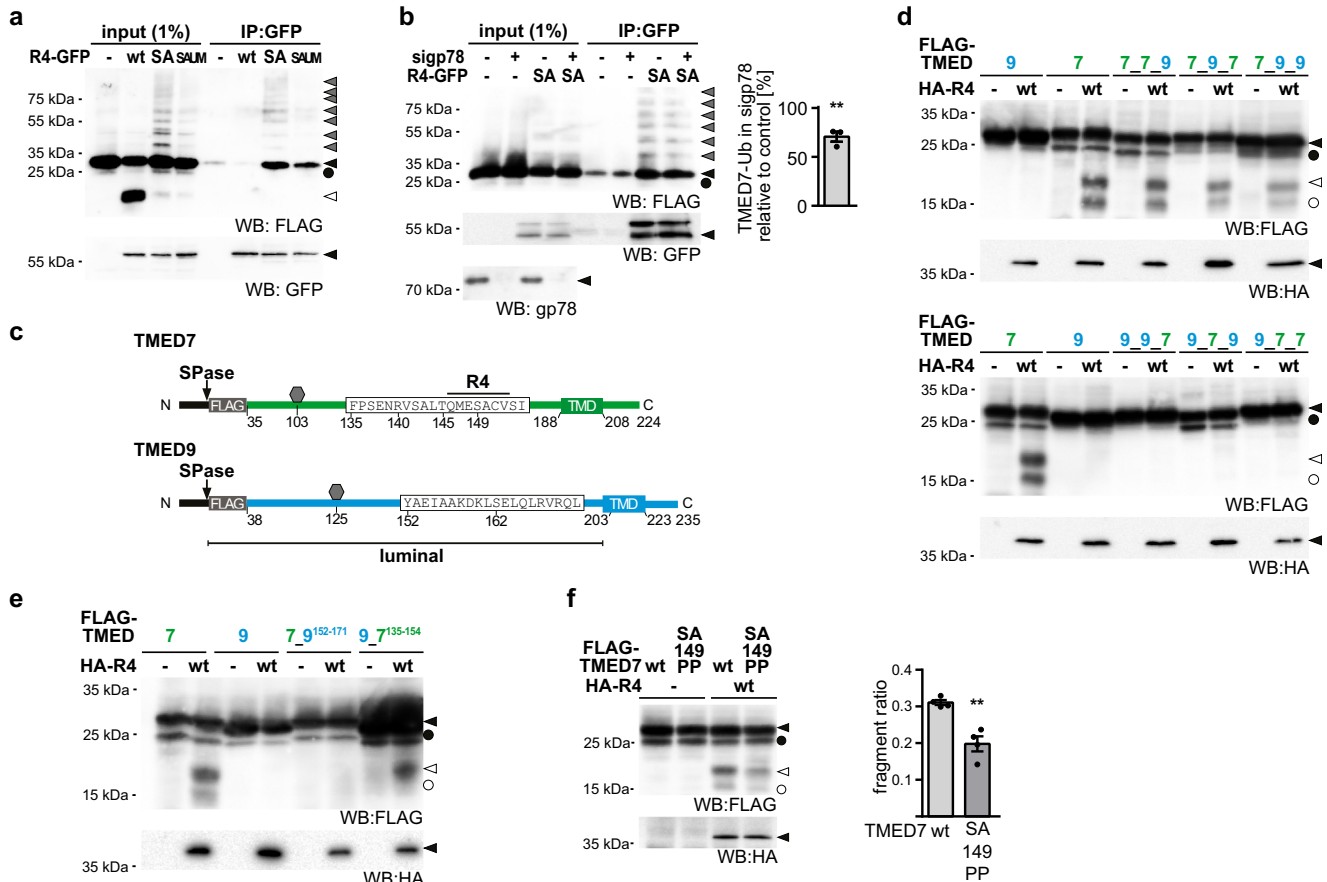

**Fig. 2 | RHBDL4 interacts with ubiquitinated TMED7 and cleavage is specified by the TMED7 luminal domain. a** HEK293T cells transiently transfected with FLAG-TMED7 and either empty vector (-), GFP-tagged RHBDL4 (R4-GFP) wt, the catalytically inactive SA mutant or the SA-UIM double mutant (SAUM) were lysed with Triton X-100 and subjected to GFP-specific immunoprecipitation (IP). Western blot (WB) analysis reveals increased co-purification of glycosylated (black arrow) and unglycosylated (circle) FLAG-TMED7, including several ubiquitinated forms (gray arrow) but not the 22-kDa cleavage fragment (open arrow) (*n* = 3). **b** HEK293T cells transfected with siRNA targeting either gp78 (sigp78) or with control siRNA (-) and either transiently transfected with FLAG-TMED7 and either empty vector (-) or the GFP-tagged RHBDL4 catalytically inactive SA mutant were lysed with Triton X-100 and subjected to GFP-specific IP. WB analysis reveals decreased co-purification of FLAG-TMED7. Quantification of ubiquitinated TMED7 in sigp78-treated cells relative to control is shown (means ± SEM, *n* = 3; **p < 0.01, two-sided Student's *t* test, *p* = 0.0043). **c** Outline of FLAG-TMED7 highlighting the 20 amino acid-cleavage site-region. Below an outline of FLAG-TMED9, including the

detailed sequence homologous to the displayed TMED7 sequence. Hexagon, position of N-linked glycan. SPase, site of predicted signal peptide cleavage; TMD, transmembrane domain. **d** HEK293T cells were transfected with either FLAG-TMED7, FLAG-TMED9 or chimera of the two. Chimera are labeled according to the following scheme, indicating the source protein for the respective part: luminal_TMD_cytoplasmic. In addition, for each substrate construct either empty vector (-) or HA-R4 was transfected. Cells were treated with MG132 (2 μM). Only for constructs with a TMED7 luminal domain a cleavage fragment (open arrow) becomes visible (*n* = 3). **e** Same experiment as in (**d**) with chimera constructs as indicated. Only for constructs containing the TMED7 amino acid sequence 135-154 a cleavage fragment (open arrow) becomes visible (*n* = 3). **f** HEK293T cells were transfected with either FLAG-TMED7 wt or a SA149PP mutant of TMED7 together with HA-R4 as indicated. Cleavage of FLAG-TMED7-SA149PP is reduced compared cleavage of wt FLAG-TMED7. Cells were treated with MG132 (2 μM), (means ± SEM, *n* = 4, **p < 0.01, two-sided Student's *t* test, *p* = 0.0019). Source data are provided as a Source Data file.

asked whether altered RHBDL4 activity manifests in changes in TLR4 signaling and inflammatory response. To study all three components of this potential signaling axis, namely RHBDL4, TMED7, and TLR4 at endogenous expression levels, we used THP-1 human monocytic cells differentiated with phorbol-12-myristat-13-acetate (PMA) to macrophage-like myeloid cells[28]. Consistent with our results in HEK293T cells, RHBDL4 knockdown in PMA-treated THP-1 cells resulted in about 50% increase in TMED7 protein levels (Fig. 3a). Because overall TLR4 levels were not significantly affected in the whole cell lysate (Fig. 3a), we assessed by flow cytometry whether RHBDL4 knockdown leads to a significant increase in TLR4 cell surface localization (Fig. 3b). Cells transfected with siRNA specific for TLR4 or TMED7 were included for comparison (Supplementary Fig. 3B, C). Consistent with a role of RHBDL4 in tuning TMED7-mediated TLR4 trafficking, we observed upon knockdown of RHBDL4 an increase of TLR4 cell surface abundance when compared to non-targeting siRNA

control (Fig. 3b). Although the effect was modest, the amplitude is in the same range of the one previously shown associated to TMED7 ablation in THP-1 cells[22]. To test the impact of the increased TLR4 cell surface abundance on downstream signaling pathways, we monitored the activation of NF-κB and induction of two cytokines whose expression is induced upon LPS binding to TLR4, namely tumor necrosis factor α (TNFα) and interleukin 6 (IL-6)[29]. Indeed, LPS-induced expression of both TNFα and IL-6 was significantly increased upon knockdown of RHBDL4, and this was TMED7-dependent, as shown by the combined knockdown of both TMED7 and RHBDL4 (Fig. 3c). This effect was corroborated by a luciferase-based reporter assay, which showed elevated LPS-induced NF-κB activation upon RHBDL4 knockdown (Supplementary Fig. 3D), whereas the UPR was not significantly activated as assessed by splicing of the XBP1 mRNA (Supplementary Fig. 3E). Consistent results were also observed with the synthetic TLR4 ligand CRX-527[30] (Supplementary Fig. 3F). Taken together these results

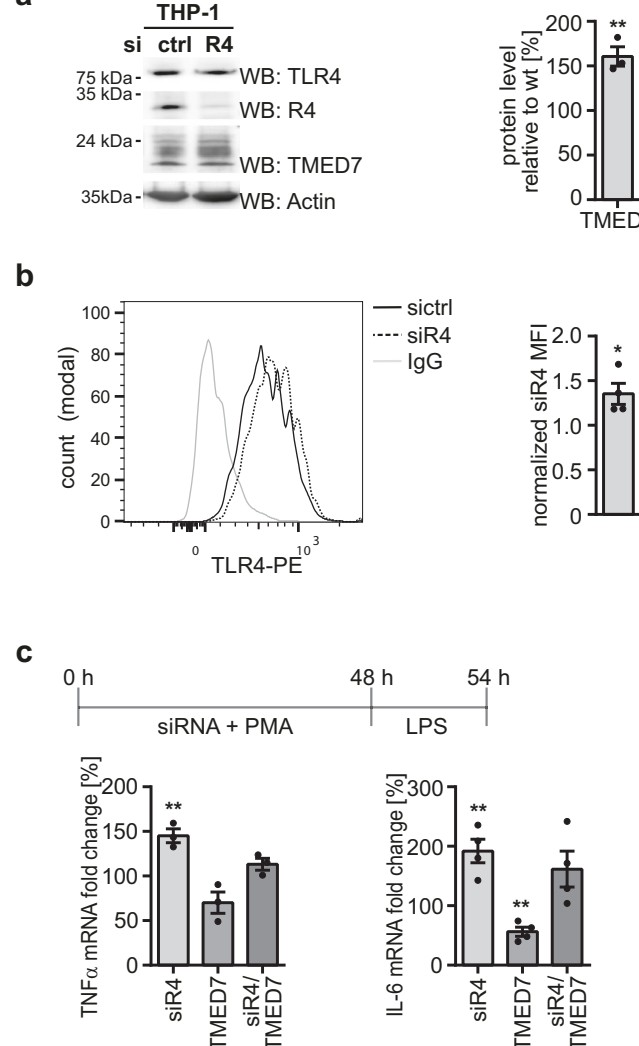

**Fig. 3 | Loss of RHBDL4 in THP-1 cells leads to TMED7 stabilization and consequently to increased TLR4 cell surface level and signaling. a** TMED7 expression is increased in THP-1 cells transfected with siRNA (si) targeting RHBDL4 (R4) relative to control siRNA (ctrl) as analyzed by western blot (WB) analysis. Actin is used as a loading control. Right panel, TMED7 quantification in siR4-treated cells relative to control (means ± SEM, $n = 3$; **$p < 0.01$, two-sided Student's $t$ test, $p = 0.005$). **b** TLR4 cell surface levels are increased in THP-1 cells transfected with siR4 compared to cells treated with sictrl. Cells have been stained with anti-TLR4-PE and analyzed by flow cytometry. Gray curve represents PE-coupled isotypic control (IgG). Representative histogram (left) and MFI ratio (right) of siR4-treated versus sictrl-treated cells. (means ± SEM, $n = 4$; *$p < 0.05$, two-sided Student's $t$ test, $p = 0.0244$). **c** THP-1 cells were transfected either with sictrl, siR4, siTMED7 or a combination of the latter two and treated with LPS (1 μg/ml) for 6 h. Normalized transcriptional levels of TNFα and IL-6 relative to LPS-treated cells. Both TNFα ($n = 3$) and IL-6 ($n = 4$) expression increases upon knockdown of RHBDL4 that depends on TMED7 expression (means ± SEM; **$p < 0.01$, two-sided Student's $t$ test, TNFα: sicon vs. siR4 $p = 0.0038$; IL-6: sicon vs. siR4 $p = 0.0036$, sicon vs. siTMED7 $p = 0.0012$). Source data are provided as a Source Data file.

show that the RHBDL4-catalyzed abundance control of TMED7 impacts TLR4 signaling on the cell surface.

Given the crucial role of TLR4 signaling in acute and chronic inflammation, we asked whether RHBDL4 modulation of TLR4 signaling also influences the inflammatory response at the organismal level. Rodents have been widely used for studying sepsis, and several models have been developed, including the administration

of LPS[24]. LPS injection in rodents results in the release of high levels of pro-inflammatory cytokines, including TNFα and IL-6, a systemic inflammatory response syndrome, and consequent dose-dependent mortality. We analyzed RHBDL4 knockout mice[31], and primary cells derived from these for their response to LPS both using in vitro and in vivo assays. Of note, RHBDL4 knockout in mice does not lead to any apparent fitness defect[31,32]. We first used bone marrow-derived macrophages (BMDMs) to study TLR4 signaling in primary cells. Consistent with our results in HEK293T and THP-1 cells, BMDMs derived from RHBDL4 knockout mice show a significant increase of the TMED7 steady-state levels compared to BMDMs from wt mice (Fig. 4a). This further resulted in a significant increase of both TNFα and IL-6 secretion in RHBDL4 ablated cells as assessed by enzyme-linked immunosorbent assay (ELISA) probing supernatant of BMDMs 6 h, 12 h, and 24 h after LPS treatment (Fig. 4b).

Next, we aimed to study how loss of RHBDL4 affects the physiological response towards LPS. It was previously shown that the susceptibility towards an endotoxin challenge correlates with cell surface abundance of TLR4[33]. RHBDL4 knockout and wt mice were given LPS intraperitoneally and 150 min later, TNFα expression induction was monitored by RT-PCR in white blood cells (Fig. 4c). Consistent with a suppressive function of RHBDL4 in TLR4 signaling, cytokine production in the knockout condition was increased. Another readout for differences in systemic inflammation is survival upon induction of septic shock. Therefore, mice were injected with a lethal dose of LPS, and the survival of the mice was monitored (Fig. 4d). RHBDL4 knockout mice showed a significantly shorter median survival: 17 h compared to 23 h for the wt mice. These results show that loss of RHBDL4 leads to an increased inflammatory response and consequently reduced survival of RHBDL4 knockout mice upon septic shock induction.

Several clinically relevant pathogens exploit TLR4-dependent signaling to trigger an innate immune response[34]. *Mycobacterium tuberculosis*-infected macrophages drive cytokine secretion via TLR4 signaling and other TLR's and genetic polymorphisms in the gene coding for this important receptor are associated with the risk for active disease after infection[35,36]. Thus, we explored whether the RHBDL4-dependent negative regulation of immune signaling is also relevant in *M. tuberculosis*-infected THP-1 macrophages. After 24 h of infection, both TNFα expression (Fig. 4e) and secretion (Fig. 4f) were significantly increased upon RHBDL4 knockdown. In addition, levels of secreted IL-1β, another cytokine that is released upon infection of macrophages with *M. tuberculosis* and triggered via TLRs[37,38], was also significantly increased in the cells mentioned above (Supplementary Fig. 4).

## Kawasaki disease-linked mutation accelerates RHBDL4 degradation

Consistent with a physiological role of RHBDL4 in controlling the innate immune response, mutation of the RHBDL4-encoding gene *Rhbdd1* has been linked to cause Kawasaki disease (GWAS ID rs139662037)[39]. Kawasaki disease is an ill-defined acute inflammatory disorder associated with elevated levels of inflammatory cytokines (including TNFα and IL-6) that affects children of young age, leading to serious vasculitis that can cause coronary artery lesions[40]. We therefore set out to test the impact of this mutation, which changed a conserved hydrophobic residue, namely isoleucine-165, located in the cytoplasmic loop 4, to a threonine (Fig. 5a and Supplementary Fig. 5A). Interestingly, expression of the RHBDL4 isoleucine-165-threonine (IT) mutant did not significantly rescue the TMED7 steady state level in RHBDL4 knockout cells when compared to the wt construct (Fig. 5b), suggesting that the missense mutation suppresses the rhomboid function. As we observed lower steady-state levels of stably expressed RHBDL4-IT, we compared its stability to the corresponding wt construct. Consistent with a partial loss-of-function phenotype, in a

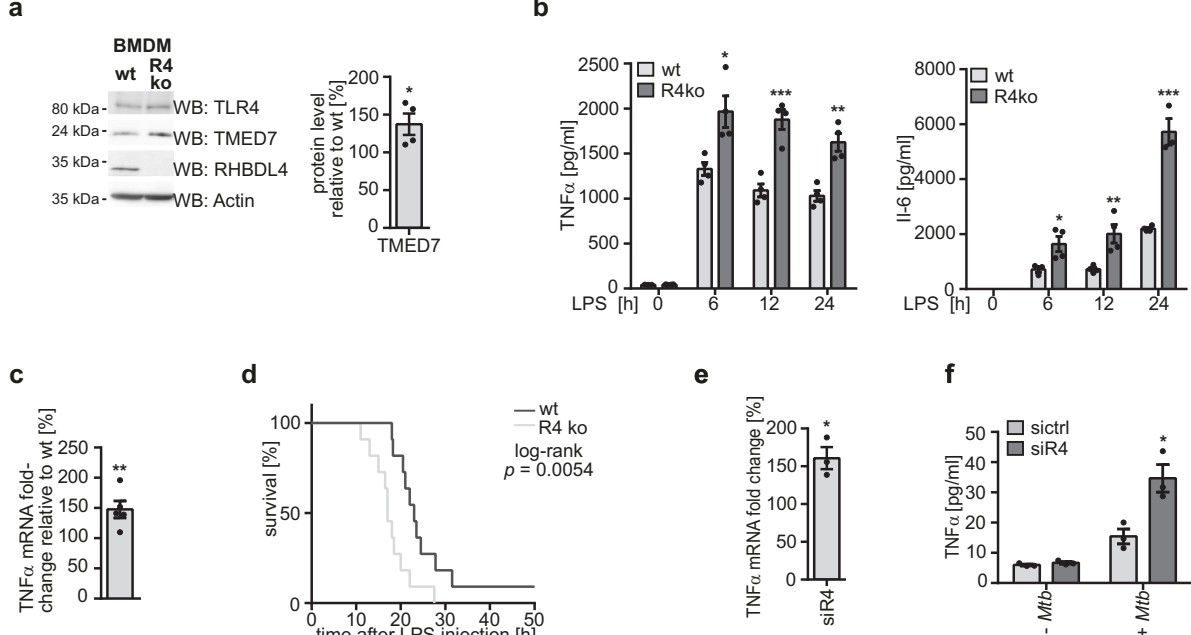

**Fig. 4 | Physiological consequence of RHBDL4 ablation. a** TMED7 expression is increased in BMDMs derived from RHBDL4 knockout (R4 ko) mice relative to cells derived from wt animals. Signal for TLR4 was obtained on a separate WB. Actin is used as a loading control. Right panel, quantification of TMED7 in R4 ko cells relative to control (means ± SEM, $n = 4$; *$p < 0.05$, two-sided Student's $t$ test, $p = 0.0383$). **b** Amount of TNFα (left) and IL-6 (right) determined by ELISA in the supernatant of wt and R4 ko BMDMs after LPS (100 ng/ml) treatment for 0, 6 (TNFα $p = 0.0154$, IL-6 $p = 0.0419$), 12 (TNFα $p = 0.001$, IL-6 $p = 0.0093$), and 24 h (TNFα $p = 0.002$, IL-6 $p = 0.0003$). Secretion of both cytokines is increased in cells derived from R4 ko mice compared to cells derived from wt mice (means ± SEM, $n = 4$ except for time points 0 h in wt animals and 24 h in R4 ko animals for IL-6 and timepoint 0 h for wt animals for TNFα that were $n = 3$; *$p < 0.05$, **$p < 0.01$, ***$p < 0.001$, two-sided Student's $t$ test). **c** Relative expression of TNFα in white blood cells from R4 ko mice is increased compared to cells derived from wt mice 150 min after injection of LPS (35 µg/g of body weight). TNFα expression has been normalized to and expressed as a ratio relative to expression in cells derived from wt mice (means ± SEM, $n = 5$ mice per condition; **$p < 0.01$, two-sided Student's $t$ test, $p = 0.0092$). **d** R4 ko mice show a decreased survival upon LPS injection (35 µg/g of body weight) when compared to wt mice ($n = 12$ mice per condition, $p = 0.0054$, Log-rank, Mantel-Cox test). **e** TNFα expression is increased in THP-1 cells upon transfection with siRNA targeting RHBDL4 (siR4) relative to control siRNA (sictrl) after infection with *M. tuberculosis* for 24 h (means ± SEM, $n = 3$, *$p < 0.05$, two-sided Student's $t$ test, $p = 0.0132$). **f** Amount of secreted TNFα determined by ELISA in the supernatant of THP-1 cells is increased in cells transfected with siRNA targeting RHBDL4 (siR4) relative to control siRNA (sictrl) upon infection with *M. tuberculosis* (+*Mtb*) for 24 h (means ± SEM, $n = 3$; *$p < 0.05$, two-sided Student's $t$ test, $p = 0.0191$). Source data are provided as a Source Data file.

cycloheximide chase experiment, the half-life of the RHBDL4-IT mutant was significantly decreased compared to wt (Fig. 5c and Supplementary Fig. 5B). Interestingly, overexpressed RHBDL4 can trigger its own degradation leading to a ladder of degradation intermediates that become visible upon longer exposure of the western blot. This processing pattern is drastically affected by the Kawasaki mutation leading to only one dominant cleavage event in the region of the first luminal loop (Fig. 5a, d). Although the etiology of Kawasaki disease remains to be investigated, this reduced protein stability of the Kawasaki missense mutation is in line with a genetic link between RHBDL4 and innate immune system control.

**Prolonged TLR4 stimulation upregulates RHBDL4 expression**

Since the TLR4-induced innate immune response has to be tightly regulated, many negative regulators of TLR4 operate in feedback loops (for review, see ref. 41). Therefore, we asked whether LPS-induced TLR4 stimulation affects RHBDL4 expression. We treated PMA-differentiated THP-1 cells and BMDMs with LPS and monitored RHBDL4 expression over time both by assessing changes in RHBDL4 mRNA and protein abundance. Indeed, in THP-1 cells 6 h after LPS addition, RHBDL4 expression was more than two-fold increased (Fig. 6a). A consistent but slightly less pronounced effect was observed in BMDMs (Fig. 6b). The elevated RHBDL4 mRNA level also resulted in increased RHBDL4 protein abundance. In BMDMs, LPS treatment caused a robust three-fold increase in RHBDL4 protein level (Fig. 6c). Likewise, starting from 12 h post LPS treatment also in THP-1 cells a subtle but significant increase of RHBDL4 protein level by about 30%

was observed (Supplementary Fig. 6). Interestingly, in mouse BMDMs LPS treatment caused the transcriptional upregulation of the UPR target BiP, whereas in THP-1, no increased BiP mRNA expression was observed (Fig. 6a). This result shows that under certain circumstances, there can be crosstalk between inflammation and control of protein homeostasis in the early secretory pathway. However, at the protein level, no significant increase in BiP expression was observed both in THP-1 cells and in BMDMs (Fig. 6c and Supplementary Fig. 6). Taken together with our previous analysis in HEK293T cells[8], these results reveal that an interplay of canonical UPR and the TLR4 signaling network transcriptionally regulates RHBDL4. While the underlying mechanism remains to be determined, we reason that RHBDL4 upregulation by TLR4 signaling serves as a negative feedback mechanism (Fig. 7).

## Discussion

Our study reveals that the ER-resident intramembrane protease RHBDL4, which forms the core of a non-canonical ERAD machinery, tunes the levels of TMED/p24 trafficking factors to control the secretion dynamics and signaling of TLR4. This hitherto uncharacterized function not only complements the physiological substrate spectrum of RHBDL4 but also reveals a surprisingly intimate connection between RHBDL4 and cargo selection and trafficking in the ER. Our data also reveal a negative feedback loop that increases RHBDL4 levels in response to LPS-induced TLR4 signaling in macrophages (Fig. 7). To our knowledge, the increased LPS sensitivity of RHBDL4 knockout mice is the first substrate-linked phenotype that has been described

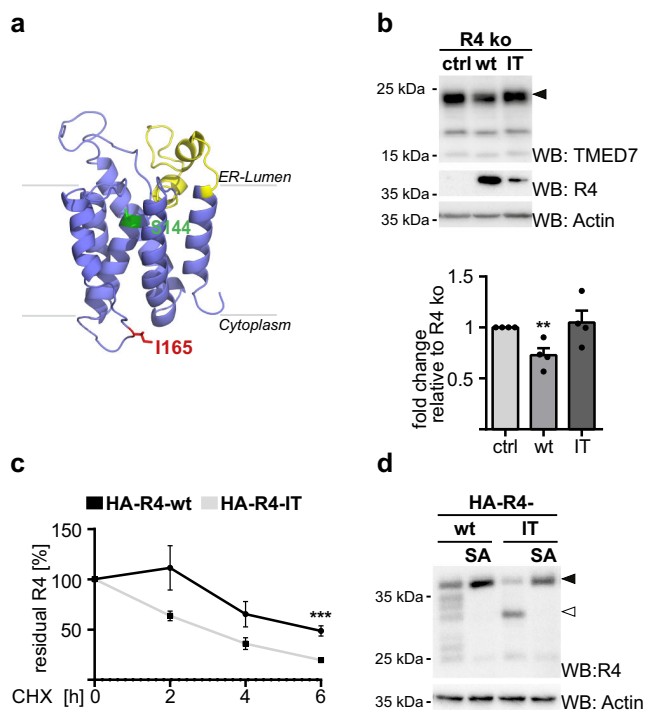

**a**

ER-Lumen

Cytoplasm

I165

**b**

R4 ko

ctrl wt IT

25 kDa -                                    ◄

15 kDa -                        WB: TMED7

35 kDa -                        WB: R4

35 kDa -                        WB: Actin

fold change relative to R4 ko

1.5

0.5

ctrl    wt    IT

**c**

■ HA-R4-wt    □ HA-R4-IT

residual R4 [%]

150

100

50

***

CHX [h] 0    2    4    6

**d**

HA-R4-

wt          IT

SA          SA

35 kDa -                              ◄

◁

25 kDa -

WB:R4

35 kDa -                        WB: Actin

**Fig. 5 | Kawasaki disease-associated mutation causes accelerated degradation of RHBDL4 by a unique autocatalytic cleavage event in its first luminal loop. a** Homology model of RHBDL4's rhomboid fold showing the position of the mutated isoleucine-165 facing the cytoplasmic surface in red. Membrane-embedded active site serine-144 and predicted main cleavage site region of the IT missense mutant are shown in green and yellow, respectively. Ribbon diagram was generated based on our previous homology model[26] by PyMol Molecular Graphics System (v.2.2.0). **b** Endogenous TMED7 levels in HEK293 T-REx RHBDL4 knockout cells (R4 ko) do not change significantly when HA-tagged RHBDL4-IT (IT) was expressed when compared to the control (ctrl) contrary to expression of the RHBDL4 wt (wt) rescue construct. Actin is used as a loading control (see Fig. 1e, means ± SEM, *n* = 4; **$p < 0.01$, two-sided Student's *t* test, ctrl vs. wt $p = 0.0075$). **c** Stability of HA-tagged RHBDL4-IT (HA-R4-IT) expressed in HEK293T cells is reduced compared to HA-tagged RHBDL4 wt (HA-R4-wt) when analyzed by cyclo-heximide (CHX) chase. Quantification of HA-tagged full-length RHBDL4 (means ± SEM, *n* = 5; ***$p < 0.001$, two-sided Student's *t* test, $p = 0.0009$). See Supplementary Fig. 5B for representative WB. **d** HA-tagged RHBDL4 wt and RHBDL4-IT expressed in R4 ko HEK293T cells are autocatalytically cleaved whereas the active site RHBDL4-SA mutant and the RHBDL4-IT-SA double mutant are stable. Actin levels were analyzed on a separate WB. (*n* = 3). Source data are provided as a Source Data file.

for organismal loss of a mammalian rhomboid protease. Although the rhomboid protease PARL has been studied in mice, the lethal multi-system phenotype observed in PARL knockouts could not be linked to one specific molecular mechanism[42–44]. Our data suggest that RHBDL4-catalyzed control of TMED7 levels plays a significant role in controlling TLR4 signaling. By interfering with TMED7 but not TLR4 itself, this enables rapid modulation of the subcellular TLR4 distribution by trafficking control without interfering with its total abundance.

TMED/p24 family proteins are trafficking factors for a diverse set of cargo, including cell surface receptors and GPI-anchored proteins[21]. As p24 proteins are thought to function as dimers or higher-order oligomers[20], their stability is expected to depend on the expression of the other family members[45,46]. TMED10 has previously been reported to be degraded by the ubiquitin-proteasome system[47], but the molecular mechanism and functional consequences have not been characterized yet. For TMED7, here we observe that RHBDL4-catalyzed cleavage can initiate its proteasomal degradation and thereby adjust its levels and consequently its ability to incorporate cargo into COPII vesicles. Our previous analysis showed that retrotranslocation of

RHBDL4-generated cleavage fragments is dependent on the RHBDL4 rhomboid-fold as well as on the recruitment of the molecular dislocase p97 to the C-terminal VBM[8,9]. In contrast to dislocation along the Hrd1-dependent ERAD route that targets full-length membrane proteins to the proteasome[3], RHBDL4-catalyzed cleavage now emerges as a potent and irreversible step that may help to fine-tune TMED/p24 protein levels. We found that of the nine human TMED/p24 proteins, only three are efficient RHBDL4 substrates, namely TMED7, −3, and −6, whereas most homologs were not substantially cleaved. Although the exact molecular mechanism of how RHBDL4 binds and selects its substrates remains to be determined (i.e., the binding constant), our results from the cell-based assay indicate that it triggers the down-regulation of a specific set of TMED/p24 proteins. Domain swap experiments show that the region of 20 amino acids surrounding the TMED7 cleavage site is sufficient for processing by RHBDL4, whereas the equivalent stretch in TMED9 is not cleaved. Although details of the determinants for RHBDL4-triggered degradation remain to be revealed, it appears that non-substrates such as TMED9 are lacking amino acids with small side chains in the corresponding sequence that are thought to facilitate rhomboid cleavage (see Supplementary Fig. 7 for sequence alignment)[9,26,48]. Moreover, we showed that TMED7 is ubiquitinated at least partially by the ERAD E3 ubiquitin ligase gp78, suggesting that cleavage occurs in concert with the ER quality control machinery. It will be interesting to see whether substrate specificity and regulation of the RHBDL4-dependent ERAD pathway show tissue-specific differences and depend on further layers of control.

Little is known about how TMED/p24 proteins select their cargo. It is commonly thought that different paralogues have distinct functions, with hetero- and homo-oligomers potentially increasing the diversity of proteins that can be recognized[20,21]. We now provide evidence that the level of selected TMED proteins is modulated by RHBDL4, indicating that induced degradation may adjust the ER export machinery to a specific need (e.g., an inflammatory program or innate immune response). However, it was previously shown that the loss of a single TMED/p24 protein leads to the destabilization of others[45,46], indicating the potential for co-regulation. This may also explain the role RHBDL4 plays in gating the ER-export of proTGFα that we previously observed[6], which may be a consequence of the TMED/p24 protein abundance changes reported here[49]. More generally, the role of proteolysis in inactivating protein function and/or serving as a degradation signal acts considerably faster than transcriptional changes, thereby ensuring the most decisive control over the proteome and signaling. It will be interesting to analyze how RHBDL4-catalyzed cleavage of TMED/p24 proteins affects protein secretion more globally. By focusing on the TMED7-mediated transport of TLR4, here we reveal an important regulatory mechanism of TLR4. However, with the identification of more cargo-receptor pairs, an even more central role in tuning secretion dynamics may become apparent.

The loss of RHBDL4 results in an increased inflammatory response and reduced survival of RHBDL4 knockout mice upon LPS injection. The primary receptor for LPS is TLR4, and therefore its biosynthesis, trafficking, signal transduction, and degradation is tightly controlled on multiple levels[25]. By controlling the trafficking of newly synthesized TLR4, TMED7 overexpression leads to increased TLR4 cell-surface expression, whereas silencing shows the opposite effect[22]. While it also has been reported that TMED7 inhibits MyD88-independent signaling upon endocytosis in endosomes[50], here we focused on the MyD88-dependent TLR4 axis at the cell surface[51]. We observed that loss of RHBDL4 leads to augmented steady-state levels of endogenous TMED7 both in HEK293T and in human and murine macrophages. Subsequently, this TMED7 increase leads to higher TLR4 cell surface levels. Upon stimulation by LPS, this results in an enhanced inflammatory response, namely a boost of TNFα and IL-6 secretion, and reduced survival of RHBDL4 knockout mice in a septic shock model. We note that a preclinical model of sepsis was out of scope of the current study

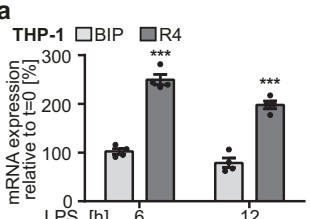 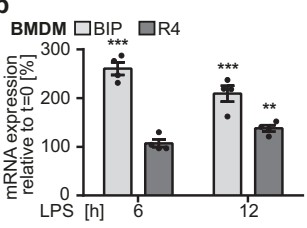 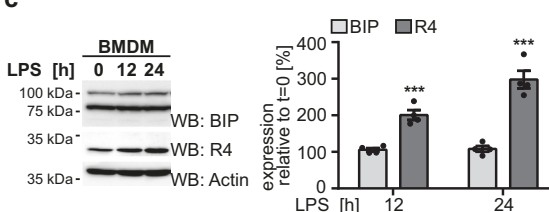

**Fig. 6 | Negative feedback regulation: TLR4 stimulation upregulates RHBDL4 expression. a** Relative mRNA expression of RHBDL4 and BiP in THP-1 cells after treatment with LPS (1 μg/ml) for the indicated time points. Expression was normalized and calculated as a ratio relative to untreated cells (means ± SEM, $n = 4$, ***$p < 0.001$, two-sided Student's $t$ test, 6 h R4: $p = <0.0001$, 12 h R4: $p = <0.0001$). **b** Same experiment as shown in (**a**) but with BMDMs treated with LPS (100 ng/ml) (means ± SEM, $n = 4$; **$p < 0.01$, ***$p < 0.001$, two-sided Student's $t$ test, 6 h BiP:

$p = <0.0001$, 12 h BiP: $p = 0.0005$, 12 h R4: $p = 0.0011$). **c** RHBDL4 protein expression increases in BMDMs after treatment with LPS (100 ng/ml) for the indicated time points as assessed by western blot (WB) analysis. BiP expression was additionally monitored. Actin is used as a loading control. Right panel, quantification of RHBDL4 and BiP expression relative to untreated control (means ± SEM, $n = 4$; ***$p < 0.001$, two-sided Student's $t$ test, 12 h R4: $p = 0.0003$, 24 h R4: $p = 0.0002$). Source data are provided as a Source Data file.

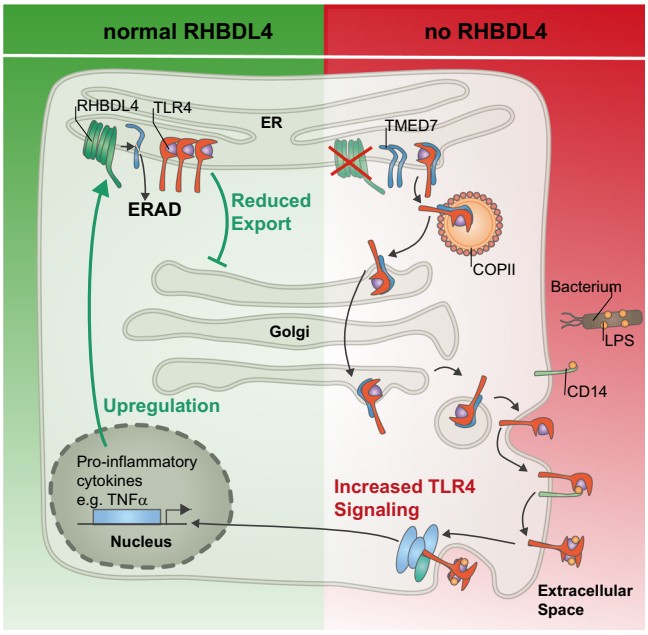

**Fig. 7 | Model of RHBDL4-mediated negative feedback regulation of TLR4 signaling.** RHBDL4-catalyzed cleavage induces downregulation of TMED7 by the ERAD pathway. Upon loss of RHBDL4, TMED7 accumulates and promotes the trafficking of TLR4 to the cell surface. Consequently, TLR4 downstream signaling is increased, resulting in enhanced cytokine secretion. As negative feedback, LPS-induced TLR4 signaling upregulates RHBDL4 expression by an unknown mechanism.

and will be needed to fully resolve the complexity of the underlying mechanism. However, survival upon LPS injection is a widely used and reliable alternative that recapitulates many characteristics of human sepsis in a highly controlled and standardized way. This effect is not only detectable upon stimulation with LPS, but also upon treatment with the synthetic TLR4 ligand CRX-527 and by infection of THP-1 cells with *M. tuberculosis* where the latter suggests that the negative regulation of the immune signaling by RHBDL4 is also clinically relevant. Consistent with a pathophysiological role of RHBDL4 as a negative regulator of the innate immune response in humans, we observed that a Kawasaki disease-associated I165T missense mutation (GWAS ID rs139662037)[39] abrogates RHBDL4 activity by triggering its auto-catalytic degradation.

By showing a TLR4-dependent transcriptional upregulation of RHBDL4, we provide evidence that RHBDL4/TMED7-mediated trafficking control is part of a negative feedback loop and contributes to

the resolution phase of TLR4 signaling (Fig. 7). LPS-induced increase of *RHBDL4* transcription was observed in THP-1 and BMDMs. This is in line with previous reports stating that TLR4 target genes are more rapidly expressed in human macrophages than in mouse macrophages following LPS exposure[52]. Interestingly, *TMED7* mRNA levels have been previously shown to fluctuate upon LPS treatment, with levels peaking at 1 h and 8 h post-LPS stimulation with a return to baseline between and after[50]. Although the functional relevance of this fluctuation needs to be further characterized, it corroborates the need for a tight regulation of TMED7 activity that RHBDL4 might further adjust on protein level. RHBDL4 was previously described to be transcriptionally upregulated during the UPR[8]. Therefore, the RHBDL4-TMED7-TLR4 axis identified here needs to be viewed in the context of the crosstalk between UPR and inflammation. Both systems share a reliance on tightly controlled signal sensing, integration of various informational cues, and the induction of cellular responses. The TLR4 receptor, for example, binds not only LPS but also endogenous products released by damaged cells. Integrating additional signals can result in responses that lead to either wound healing or tissue damage[53]. The same holds true for the UPR, which can either restore protein-folding homeostasis or, if not successful, promote apoptosis[54]. RHBDL4 is bi-directionally connected to these two highly complex cellular signaling networks and might be involved in integrating both. Further studies will be needed to understand the molecular regulation of RHBDL4 in both pathways and the relative impact RHBDL4 has on initiating a tuned cellular response. This has therapeutic implications as UPR-associated inflammatory pathways are involved in inflammatory diseases such as diabetes, inflammatory bowel diseases, and cancer[55].

## Methods

### Approval

Experiments with mice were performed in accordance with protocols approved by the Ethics Committee of the Instituto Gulbenkian de Ciência and the Portuguese National Entity Direção Geral de Alimentação e Veterinária (DGAV) and with the Portuguese (Decreto-Lei no.113/2013) and European (directive 2010/63/EU) legislation related to housing, husbandry, and animal welfare.

### Plasmids

All constructs were cloned into pcDNA3.1(+) (Invitrogen). Human RHBDL4 (Gene ID 84236 IMAGE 40023929 [https://www.ncbi.nlm.nih.gov/nuccore/71682850]) was cloned with an N-terminal triple HA-tag or a C-terminal GFP-tag. RHBDL4 mutants were generated by site-directed mutagenesis, mutating serine-144 to alanine (RHBDL4-SA), and additionally leucine-274 and leucine-278 to alanine (RHBDL4-SA-UIM) or isoleucine 165 to threonine (RHBDL4-IT) and additionally mutating serine-144 to alanine (RHBDL4-IT-SA). TMED2 (Gene ID

10959, full-ORF Gateway cDNA clone GenBank accession CR541682.1), TMED3 (Gene ID Hs.513058, full-ORF Gateway cDNA clone GenBank accession CV023374.1), TMED5 (Gene ID 50999, full-ORF Gateway cDNA clone GenBank accession LT736049.1), TMED6 (Gene ID 146456, full-ORF Gateway cDNA clone GenBank accession LT736254.1), TMED7 (Gene ID 51014, full-ORF Gateway cDNA clone GenBank accession LT744922.1), TMED9 (Gene ID 54732, full-ORF Gateway cDNA clone GenBank accession LT736220.1), and TMED10 (Gene ID 10972, full-ORF Gateway cDNA clone GenBank accession JF432606) were amplified without their signal sequence by PCR and sub-cloned into a pcDNA3.1-based expression vector introducing an N-terminal triple FLAG-tag downstream of a signal sequence[27]. Open reading frames for TMED1 (Gene ID 11018, reference sequence NM_006858.4), TMED4 (Gene ID 222068, reference sequence NM_001303058.1), and p24 chimera were ordered as synthetic genes without their signal sequence (Integrated DNA Technologies, Coralville, US) and sub-cloned into pcDNA3.1 accordingly. For TMED7, the following three fragments were used: residues 35-187 as luminal, residues 188-208 as TM domain, and residues 209-224 as cytoplasmic. For TMED9, the following three fragments were used: residues 38-202 as luminal, residues 203-223 as TM domain, and residues 224-235 as cytoplasmic. For the 20 amino acid cleavage site chimera, residues 135-154 of TMED7 were replaced by residues 152-171 of TMED9 and vice versa. TMED7 reference peptides were generated by the introduction of a premature stop codon at the indicated positions by site-directed mutagenesis. Accordingly, the TMED7-SA149PP mutant was generated by site-directed mutagenesis.

## Culture of cell lines

HEK293T cells (CRL-3216, ATCC) and RHBDL4 knockout cells[9] were grown in DMEM (Gibco) supplemented with 10% fetal bovine serum (FBS) at 37 °C in 5% $CO_2$. To produce inducible stable transfected HEK293 T-REx Flp-In wt (R78007, Thermo Scientific) and RHBDL4 knockout cells, the cells were transfected with a plasmid encoding Cas9 and a single guide RNA targeting exon 4 in the *Rhbdd1/RHBDL4* gene (fwd: caccgGTTGAGGGCCAAAGTTGCTA, rvs: aaacTAG-CAACTTTGGCCCTCAACc) and selected with puromycin (3 µg/ml) for 120 h. For stable transfection with either an empty vector or a plasmid expressing triple HA-tagged RHBDL4-wt, RHBDL4-IT, or RHBDL4-SA, pcDNA5/FRT/TO constructs were co-transfected with pOG44 (Invitrogen) followed by selection with blasticidin (10 µg/ml) and hygromycin B (125 µg/ml) in DMEM supplemented with 10% FBS containing 1% GlutaMAX (Gibco) and 1% sodium pyruvate (Gibco). Expression was induced with 1 µg/ml doxycycline for 24 h–48 h. THP-1 cells (gift from G. Stoecklin, Mannheim University) were grown in T-175 cell culture flasks with a ventilation cap in RPMI 1640 medium supplemented with 10% FBS and 50 µM 2-mercaptoethanol. To chemically induce monocytic differentiation in THP-1 cells, 500 nM PMA (Sigma-Aldrich) was added for 48 h. In the case of THP-1 cells, the PMA-containing medium was replaced by fresh culture medium supplemented with 1 µg/ml LPS (from *Escherichia coli* O55:B5, Sigma-Aldrich) or 0.1 µg/ml CRX-527 (Invivogen) for the indicated periods of time. For the NF-κB luciferase assay, THP1-Lucia NF-κB cells (Invivogen, thpl-nfkb) were grown in RPMI 1640 medium containing 10% heat-inactivated FBS, 2 mM L-glutamine, 25 mM HEPES, 100 µg/ml Normocin and 100 µg/ml Pen/Strep. After siRNA transfection, cells were treated with 500 nM PMA for 24 h, followed by growth in normal medium for 48 h, before treatment with 0.1 µg/ml LPS for 24 h. Luciferase activity was analyzed by luminescence assays using an Infinite M1000 plate reader (Tecan) with the SPARKCONTROL software (version 3.2) according to the manufacturer's instructions. All used cell lines were regularly screened for mycoplasma contaminations.

## Culture of primary cells

We generated BMDMs from one adult wt and one RHBDL4 knockout mouse. The femurs were cleaned from fur and muscles and transferred to a sterile dish containing RPMI 1640 medium. The femurs were cut open, and the bone marrow was flushed into serum-free RPMI medium and transferred to a 15 ml tube. The bone marrow cells were pelleted at 2000 x g for 5 min and resuspended in 30 ml RPMI 1640 medium supplemented with 10% (v/v) FBS, 1% GlutaMAX, 1% (v/v) penicillin/streptomycin, gentamycin (10 µg/ml), 2-mercaptoethanol (50 µM) and 20% of filtered conditioned culture supernatant from confluent L929 cells (ATCC CCL-1). Three days after isolation, the cells were covered with an additional 10 mL of fresh culture medium. Subsequently, the medium was replaced every second day until the cells were used. When confluent or before plating for an experiment, cells were washed with PBS, incubated in PBS for 10 min at 4 °C and detached with a cell scraper. In general, cells were plated for experiments 8-11 days after extraction.

## Transfection

To achieve specific knockdown of TMED7, TLR4, RHBDL4, and gp78 cells were transfected with 50 pmol ON-TARGETplus SMARTpool human siRNA (Horizon Discovery, RHBDL4: L-019378-00-0005, sequences: 5'-CGGCAAUACUACUUUAAUA-3', 5'-CGAGGAAAUACCAG AAAUA-3', 5'-GGGAUAAAUACUGGACUUA-3', 5'-GACAGCGGCUUCAC AGAUU-3', TMED7: L-007855-02-0005, sequences: 5'-CAGCAUGGGC AUAUGUAAA-3', 5'-ACACAUAAGUGCCAUACAU-3', 5'-UCGCAGUUGU UUAUAUCUA-3', 5'-GAAGUUGUCUUGCGGCUUU-3', TLR4: L-008088-01-0005, sequences: 5'-UGGUGGAAGUUGAACGAAU-3', 5'-GUUUA-GAAGUCCAUCGUUU-3', 5'-CAUUGAAGAAUUCCGAUUA-3', 5'-GGA AAAUGGUGUAGCCGUU-3') per well of a 12-well plate using RNAiMAX (Thermo Fisher Scientific). As control, the same amount of scrambled control siRNA (Horizon Discovery, D-001810-10-20, sequences: 5'-UGGUUUACAUGUCGACUAA-3', 5'-UGGUUUACAUGUUGUGUGA-3', 5'-UGGUUUACAUGUUUUCUGA-3', 5'-UGGUUUACAUGUUUUCCUA-3') was transfected. HEK293T cells were transfected 24 h after seeding, while THP-1 were transfected simultaneously with seeding and PMA addition. Transient transfections with plasmid DNA were performed using 25 kDa linear polyethyleneimine (Polysciences)[56]. Generally, 500 ng DNA were transfected for all substrate candidates with 200 ng plasmid encoding RHBDL4 while adjusting the total DNA amount per condition to 2 µg using empty vector DNA in a 6-well format. The amounts were adjusted to other formats in relation to the surface area. Proteasomal degradation or the AAA-ATPase p97 was inhibited by treating the cells with 2 µM MG132 (Calbiochem) or 1 µM CB-5083 (ApexBio) for 16 h prior to harvest.

***M. tuberculosis* cultivation and THP-1 infection experiment.** For all infection experiments the mycobacterial strain Erdman was used and grown in Middlebrook 7H9 broth (Becton Dickinson) supplemented with 0.2% glycerol (Th. Geyer GmbH & Co. KG), 10% albumin dextrose catalase (ADC Middlebrook; Becton Dickinson) and 0.05% Tween80 (Carl Roth). $5*10^5$ PMA-differentiated THP-1 (PMA differentiation is described above) cells were infected with *Mtb* Erdman. Medium was replaced prior to infection and *M. tuberculosis* was added with a multiplicity of infection of 5. After 24 h post infection, supernatants were collected and RNA was isolated using RNeasy mini kit (Qiagen). Supernatants and freshly isolated RNA were stored at −20 °C.

## Immunoprecipitation

Cells were solubilized in solubilization buffer (50 mM HEPES-KOH, pH 7.4, 150 mM NaCl, 2 mM MgOAc₂, 10% Glycerol, 1 mM EGTA, 10 mM N-ethylmaleimide) supplemented with 1% Triton X-100, EDTA-free complete protease inhibitor cocktail (Roche) and 10 µg/ml phenylmethylsulfonyl fluoride (PMSF). After lysis on ice for 30 min, the nuclear fraction was separated by centrifugation at 16,000 x g for 15 min at 4 °C. The supernatant was pre-cleared with protein G-coated sepharose beads (GE Healthcare). For anti-FLAG immunoprecipitation, the precleared lysate was incubated with anti-FLAG M2

agarose (Sigma). For anti-GFP immunoprecipitation, the precleared lysate was incubated with 1.6 mg anti-GFP antibody (Roche), followed by incubation with protein G beads. The beads were washed thrice with lysis buffer containing 0.5% Triton X-100. Proteins were eluted from the beads by incubating with SDS-sample buffer and analyzed by western blotting.

## Cycloheximide chase analysis

48 h after transfection, Hek293T cells were treated with 100 µg/ml cycloheximide, harvested after indicated time points and analyzed by western blot analysis as described below.

## Cellular fractionation and proteomic analysis

Cells were cultured in DMEM medium (Silantes) supplemented with 10% dialyzed FBS (Silantes), and either light arginine and lysine or heavy Lys-8 ($^{13}C_6$, $^{15}N_2$), Arg-10 ($^{13}C_6$, $^{15}N_4$) isotopes for approximately six doublings. The samples were pooled and microsomes were prepared by swelling the cells in hypotonic buffer (10 mM HEPES-KOH, pH 7.4, 1.5 mM MgCl$_2$, 10 mM KCl, 0.5 mM DTT) containing 10 µg/ml PMSF and EDTA-free complete protease inhibitor cocktail (Roche) and lysed by passing five times through a 27-gauge needle. After the removal of cell debris, the supernatant was subjected to centrifugation for 20 min at 4 °C by 100,000 x g. The pellet was resuspended in rough microsome buffer (50 mM HEPES pH 7.4, 250 mM sucrose, 50 mM KOAc, 2 mM Mg(OAc)$_2$, 1 mM DTT), and extracted by 500 mM KOAc and 50 mM EDTA for 15 min on ice followed by centrifugation through a sucrose cushion (500 mM sucrose, 50 mM HEPES, 500 mM KOAc, 5 mM Mg(OAc)$_2$) for 45 min at 140,000 × g and 4 °C. Subsequently, the pellet was resuspended in a freshly prepared ice-cold sodium carbonate solution (100 mM) and transferred onto an alkaline sucrose cushion (125 mM sucrose, 100 mM Na$_2$CO$_3$) and centrifuged for 45 min at 140,000 × g and 4 °C. The cell pellet was resuspended in rough microsome buffer with the help of a Dounce homogenizer. For proteomic analysis, protein samples were dissolved in SDS-sample buffer (see below) and separated by SDS-PAGE for 1 cm. Gels were stained with Quick Coomassie Stain (Serva). Each gel lane was cut into two pieces, washed with 50% acetonitrile, reduced with DTT and alkylated with iodoacetamide, followed by digestion with trypsin in 0.01% trifluoroacetic acid. Tryptic peptides were extracted with 50% acetonitrile and 10% formic acid. Acetonitrile was removed, and peptides were mixed with 1% trifluoroacetic acid.

## MS analysis

The samples were analyzed by a UPLC system (nanoAcquity) coupled to an ESI LTQ Orbitrap mass spectrometer (Thermo Scientific). The uninterpreted MS/MS spectra were searched against the SwissProt-human database using MaxQuant software. The algorithm was set to assuming acetylation, deamidation and oxidation as variable modifications of peptides with a minimal length of seven amino acids. For further analysis, the Perseus software (version 1.6.14.0) was used. Rows "only identified by site", "reverse" and "potential contaminants" were removed. Proteins were filtered for GOCC term "ER" and H/L ratio of equal or greater than 1.0. Data from two independent experiments were combined for Supplementary Data 1.

## Flow cytometry

For TLR4 cell surface analysis, $5 \times 10^5$ cells were seeded and transfected with siRNA, as described above. 48 h after transfection, the cells were washed with PBS and incubated in PBS on ice for 10 min. Cells were detached, spun down and resuspended in 50 µl PBS with 5% (v/v) FBS. 5 µl Fc Block (BD Biosciences) were added, and cells were incubated on ice for 25 min. Afterwards, the cells were stained with 10 µl phycoerythrin (PE)-coupled anti-TLR4 antibody or PE-coupled isotypic control for 20 min. Cells were washed once with PBS and resuspended in 300 µl PBS containing 5% (v/v) FBS. Cells were analyzed on a

FACSCanto II (BD Biosciences) using the BD FACSDiva software (v.8.0.2). Forward light scatter (FSC), side light scatter (SSC), and fluorescence emission after excitation with 488 nm were acquired. The samples were analyzed using a FlowJo software, version 10.2.

## Mouse experiments

All mice used in this study were crossed in the C57BL/6 J background. The RHBDL4 knockout mouse was a generous gift by M. Freeman (Cambridge/Oxford, UK) and had been described previously[31]. All in vivo experiments were carried out after co-housing of wt and RHBDL4 knockout mice for at least eight weeks to normalize potential differences in microbiota. For male mice, bedding was regularly intermixed between cages of wt and RHBDL4 knockout mice.

## LPS challenge and blood collection

35 µg LPS per g of mouse weight were administered to male wt or RHBDL4 knockout mice of approximately 300 days of age by intraperitoneal injection. After 150 min, these mice were sacrificed using CO$_2$, and the blood was collected by cardiac puncture using a 1 ml syringe containing 200 µl 0.5 M EDTA. The blood samples were collected into 1.5 ml tubes and centrifuged at 1000 × g for 10 min at room temperature. Plasma was collected and frozen at −20 °C. For isolation of the white blood cells, the white layer covering the erythrocyte fraction was removed and transferred into a 15 ml tube containing 10 ml of red blood cell lysis buffer (NH$_4$Cl 8.02%, NaHCO$_2$ 0.84%, EDTA 0.37% in MQ water). The tubes were inverted several times, followed by incubation at room temperature for 10 min to let the red blood cell lysis proceed. The samples were centrifuged at 1000 × g for 5 min, the supernatant was discarded, and the blood lysis step was repeated by adding additional 7 ml of red blood cell lysis buffer. After 5 min, lysis was stopped by adding 7 ml PBS, the tubes were centrifuged at 1000 × g for 5 min, and the supernatant was removed. The RNA was extracted using the RNA isolation kit (Macherey-Nagel).

## Survival after LPS challenge

Female WT or RHBDL4 knockout mice of approximately 220 days were administered 35 µg/g of body weight LPS resuspended in PBS by intraperitoneal injection. The survival of the mice was monitored for 50 h by observing mice at least every 90 min.

## ELISA

Conditioned medium of LPS-treated $2 \times 10^6$ BMDMs or $5*10^5$ M. tuberculosis- infected THP-1 cells was harvested. Following the manufacturer's instructions of ELISA kits specific for IL-6, TNFα or IL-1β (Invitrogen) the concentration of the cytokines was determined. Measurements were made with the M1000Pro plate reader (Tecan) using Tecan i-control software (v1.10.4.0) or an Hidex Sense microplate reader (Hidex) and were performed in technical duplicates. A four-parameter logistic curve fit was plotted to the standards, and the concentration of the samples was calculated accordingly.

## Antibodies

The following antibodies were used with indicated dilutions: 1:4000 mouse monoclonal anti-β-actin (clone AC-15, A1978, Sigma), 1:100 rabbit anti-BiP (ab21685, Abcam), 1:1000 mouse monoclonal anti-FLAG-HRP (clone M2, A8592, Sigma), 1:1000 mouse monoclonal anti-GFP (clones: 7.1 and 13.1, 11814460001, Roche), 1:1000 mouse monoclonal anti-gp78 (clone 3D9, H00000267-M01, Novus Biologicals), 1:1000 mouse monoclonal anti-HA (clone 16B12, 901502, Biolegend), 1:500 rat monoclonal anti-HA (clone 3F10, 11867431001, Roche), 1:250 rabbit anti-RHBDL4 (HPA013972, Sigma), 1:1000 rabbit anti-TMED7 (gift from F. Wieland, Heidelberg University), 1:1000 rabbit anti-TMED2 (gift from F. Wieland, Heidelberg University), 1:1000 rabbit anti-TMED10 (gift from F. Wieland, Heidelberg University), 1:100 mouse monoclonal anti-TLR4 (clone: 25, sc-293072, Santa Cruz

Biotechnology), 1:20 mouse monoclonal anti-TLR4-PE (clone: HTA125, sc-13593, Santa Cruz Biotechnology), 1:2000 mouse monoclonal anti-ubiquitin (clone P4D1, sc-8017, Santa Cruz Biotechnology).

## Western blotting
HEK293T cell lysates were prepared in SDS-sample buffer (50 mM TRIS-pH 6.8, 10 mM EDTA pH 8.0, 5% glycerol, 2% SDS, 0.01% bromophenol blue) containing 5% 2-mercaptoethanol. THP-1 cell lysates were prepared by solubilizing the cells for 10 min in Triton X-100 lysis buffer (20 mM HEPES pH 7.4, 150 mM sodium chloride, 1.5 mM magnesium chloride, 1 mM EGTA, 10% (v/v) glycerol, 10 μg/ml PMSF, 1% (v/v) Triton X-100, 1x Roche protease inhibitor cocktail) on ice. The samples were centrifuged at $16,000 \times g$ for 15 min, and the soluble fraction was mixed with SDS sample buffer. For western blot analysis, samples were incubated for 10–15 min at 65 °C, resolved on Tris-glycine SDS-polyacrylamide gels and transferred onto PVDF membrane. After antibody exposure, enhanced chemiluminescence analysis was performed to detect antibody signals. For detection the LAS-4000 system (Fuji) with the ImageQuant LAS 4000 software and ImageQuant 800 Fluor system (Cytiva) with the ImageQuant 800 software (version 2.0.0) were used. Images were analyzed using Fiji[57].

## Real-time quantitative reverse transcription and polymerase chain reaction
RNA was isolated using the NucleoSpin RNA kit (Macherey-Nagel) according to the manufacturer's instructions. cDNA was synthesized using the RevertAid First Strand cDNA Synthesis Kit (Thermo Fisher) with the accompanied Random Hexamer Primers according to the manufacturer's instructions. Quantitative PCR (qPCR) reactions were performed using the SensiFAST SYBR No-ROX Kit (Bioline) according to the manufacturer's instructions in a 384-well format in technical triplicates. Primers targeting human actin (fwd: GCATTGCCGA-CAGGATGC, rvs: GCAATGATCTTGATCTTCATTGTGC), mouse actin (fwd: CATTGCTGACAGGATGCAGAAGG, rvs: TGCTGGAAGGTGGA-CAGTGAGG), mouse TATA-binding protein (fwd: CCTTCACCAAT-GACTCCTATGACC, rvs: TATTTTGAAGCTGCGGTACAATTCC), human beta-2-microglobulin (fwd: CACGTCATCCAGCAGAGAAT, rvs: TGCT GCTTACATGTCTCGAT), human BiP (fwd: CCAACGCCAAGCAAC-CAAAG, rvs: TGCCGTAGGCTCGTTGATG), mouse BiP (fwd: TCGA-TACTGGCCGAGACAAC, rvs: CGACGGTTCTGGTCTCACAC), human RHBDL4 (fwd: GGTCGTAGAGAGCGTTCAGC, rvs: CTTGATCTCCGTTG CATGGC), mouse RHBDL4 (fwd: GGGCCTCTGAAGAAAATCATGG, rvs: GGGCCTGCATTGTTAAAGTGG), human TMED2 (fwd: AGCACGAA-CAGGAATACATGG, rvs: TCATGGCAACTAGAACAAGAG), human TME D10 (fwd: GCGGATACCTGACCAACTCG, rvs: TCGCAGCTCTACCTC-TAATGG), human TMED7 (fwd: CTTCGAGCTTCCTGACAACG, rvs: ATCGACAATCTACATCATAGTGACC), human TNFα (fwd: CAGCCTCT TCTCCTTCCTGAT, rvs: GCCAGAGGGCTGATTAGAGA), mouse TNFα (fwd: GGACTCAAATGGGCTTTCCG, rvs: GAGACAGAGGCAAC CTGACC), human IL-6 (fwd: AATTCGGTACATCCTCGACGG, rvs: GGTTGTTTTCTGCCAGTGCC), and mouse IL-6 (fwd: AGAC TGGGGATGTCTGTAGC, rvs: CAACTGGATGGAAGTCTCTTGC) were used. qPCRs were run on the Roche Light Cycler 480 I or II using the Light Cycler 480 SW software (version 1.51). Relative changes in gene expression were calculated using the 2^CCt method. In HEK293T cells, expression of beta-2-microglobulin and TATA-binding protein were used for normalization, while in THP-1 cells, beta-actin was used instead of beta-2-microglobulin. For the *M. tuberculosis* infection experiment and the expression analysis in BMDMs, only TATA-binding protein was used for normalization.

As UPR assay, cDNA was used as template for PCR amplification across the fragment of the XBP1 cDNA bearing the intron target of IRE1α ribonuclease activity[58]. Primers used were 5′-CCTGGTTGCTGAAGAGGAG and 5′-CCATGGGGAGATGTTCTGG, leading to a 145 bp amplicon from unspliced XBP1 and a 119 bp amplicon

from spliced XBP1. PCR conditions were: 95 °C for 5 min; 95 °C for 1 min; 50 °C for 1 min; 72 °C for 45 s; 72 °C for 5 min with 40 cycles of amplification. PCR products were resolved on a 2.5% agarose/1x TAE gel and stained with ethidium bromide. Agarose gels were imaged with the BioRadGel Doc EZ Imager using the Image Lab software (version 6.1).

## Quantification and statistical analysis
If not stated otherwise assays were conducted in triplicates and measurements were taken from distinct samples; where indicated unpaired two tailed Student's $t$ test was performed. Statistical parameters are given in the source data file. Survival data was analyzed by log-rank test on a Kaplan-Meier survival curve. Data are provided with +/− standard error of the mean. $*p < 0.05$, $**p < 0.01$, $***p < 0.005$. For quantification of cleavage efficiency, the amount of cleavage product was normalized to the amount of full-length protein. Statistical analysis was performed using GraphPad (version 6) for Windows (GraphPad Software). Analysis of qPCR, ELISA, luminescence assay, and western blots was performed in Microsoft Office 2016 Excel.

## Reporting summary
Further information on research design is available in the Nature Portfolio Reporting Summary linked to this article.

## Data availability
The authors declare that the data supporting the findings of this study are available within the paper and its supplementary information files. Source data are provided in the Source data file. The mass spectrometry proteomics data have been deposited to the ProteomeXchange Consortium via the PRIDE[59] partner repository with the dataset identifier PXD045934. Source data are provided with this paper.

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

## Acknowledgements

RHBDL4 knockout mice were a generous gift from Matthew Freeman and the MRC Laboratory of Molecular Biology Cambridge (UK). We thank Érika de Carvalho, Emma Burbridge, Miguel Cavadas, Nieves Peltzer, Michael Dal Molin, and Jason Chhen for technical assistance and helpful comments, Thomas Ruppert (ZMBH MS facility) for the MS analysis, and Nick Orr for helpful suggestions to the GWAS analysis. The work was funded by the Deutsche For-schungsgemeinschaft (DFG, German Research Foundation) – Project-ID 201348542 – CRC 1036/TP12, the Center of Molecular Medicine Cologne (CMMC) – Project-ID C10, and a fellowship of the Boehringer Ingelheim Fonds to J.D.K. J.R. is supported by the German Research Foundation (DFG) grant SFB 1403, the German Center for Infection Research (DZIF; TTU-TB grants 02.806, 02.814 and 02.913), the German Federal Ministry of Education and Research BMBF, grant IdEpiCo and by the European Union Inno-vative Medicines Initiative 2 Joint Undertaking program grant no. 853989 (ERA4TB). S.J.T. and J.R. are supported by a research grant of the CMMC (B10), and S.J.T. by stipends from the Imhoff-Stiftung and the Köln Fortune Program.

## Author contributions

JDK and MKL conceived the study. JDK and SSS led the experimental work and together with FK, and MT performed most experiments. NK and JDK performed the cellular fractionation and proteomic analysis. SJT and SSS performed the M. tuberculosis infection experiment. MB helped with the mouse experiments. Data was analyzed by JDK, SSS, FK, SJT, JR, CA, and MKL. The paper was written by JDK and MKL with contribution from all other authors.

## Funding

## Competing interests

The authors declare no competing interests.
