## [Peer Review File · Nature Communications]

RHBDL4-triggered downregulation of COPII adaptor protein
TMED7 suppresses TLR4-mediated inflammatory signalingREVIEWER COMMENTS

Reviewer #1 (Remarks to the Author):

In the manuscript “RHBDL4-mediated downregulation of COPII adaptor protein TMED7 suppresses TLR4-mediated inflammatory signaling”, Knopf et al. described their discovery that RHBDL4-mediated degradation of the p24-family cargo receptor, TMED7, decreases TLR4 density on the plasma membrane. Furthermore, they identified that immune response triggered by lipopolysaccharide upregulates RHBDL4, which results in a negative feedback loop in decreasing the TLR4 density, reducing TLR4-dependent inflammation response. The mechanistic model was further validated in mouse model.

The manuscript is well-written with a carefully mapped pathway, and provides new insight into the functional importance of cargo receptors. We have just a few minor points that should be addressed:

1. In Fig. 3B, the changes in flow cytometry with siRNA treatment are modest, as the authors pointed out. It would be helpful if the authors can provide associated quantification of the data (e.g., number of cells in each replica and median/CV of the population distribution) to help the reader with the visualisation.
2. As the total abundance of TLR4 is not affected by down regulation of TMED7, it might be interesting to see where the excess TLR4 localizes -- will this cause extra stress on the cells?
3. The specificity of RHBDL4-mediated degradation of the p24 paralogs is remarkable. Does the specificity correlate with sequence similarities among the p24 proteins? It would be interesting to have a brief alignment of TMED sequences in the domain recognized by RHBDL4 (similar to that mapped by TMED9 domain swapping by the authors) to address this point in the discussion section.
4. Some minor typos surrounding Figure S1: Figure S1B doesn't have all the p24s, and the panel doesn't match the figure legend with regard to labelling; line 146, should refer to S1C; line 151 should refer to S1D.

Reviewer #2 (Remarks to the Author):

In this study, the authors demonstrate that RHBDL4 is a new negative regulator of TLR4 by targeting and degrading TMED7, a member of the p24 family of COPII adaptor proteins, which counteracts the transport of TLR4 to the cell surface. The study primarily focused on an in vitro setting using human HEK293 cells, THP1 cells, and murine bone marrow-derived macrophages. However, the findings could be strengthened if the authors also included a pre-clinical model of sepsis instead of endotoxemia, as this would be a more relevant and clinically meaningful model. Additionally, the lack of an LD50 survival model is a limitation of the study. The authors did not mention the impact of this pathway utilizing other ligands of TLR4, which could be important to consider. To better understand the compartmentalization and trafficking of TLR4, the authors could have provided confocal microscopy data. It would also have been informative to assess TLR4-associated adaptor molecules and the transcription factor NF- κ B or AP-1 when establishing RHBDL4 as a negative regulator of TLR4.

We very much thank the reviewers and editor for the helpful suggestions. Please find below a reply to all the constructive critiques on our study. Moreover, we have intensified our effort to support the biological relevance of the RHBDL4-mediated control of innate immune responses as follows.

In support of our model, we observed that a missense mutation that is linked to Kawasaki syndrome, an ill-defined inflammatory disease in children, leads to a significant destabilization of RHBDL4 by an unprecedented acceleration of autocleavage. We have added this set of experiments in totally new Figure 5 and S5. To our knowledge, this is the first known patient mutation in a secretory pathway rhomboid, which potentially opens a new avenue in the understanding of the Kawasaki syndrome.

Moreover, we have added new data on an in vitro Tuberculosis THP-1 cell infection model (Figure 4E-F and S4), which underlines the medical relevance of our findings.

Reviewer #1

In the manuscript “RHBDL4-mediated downregulation of COPII adaptor protein TMED7 suppresses TLR4-mediated inflammatory signaling”, Knopf et al. described their discovery that RHBDL4-mediated degradation of the p24-family cargo receptor, TMED7, decreases TLR4 density on the plasma membrane. Furthermore, they identified that immune response triggered by lipopolysaccharide upregulates RHBDL4, which results in a negative feedback loop in decreasing the TLR4 density, reducing TLR4-dependent inflammation response. The mechanistic model was further validated in mouse model.

The manuscript is well-written with a carefully mapped pathway, and provides new insight into the functional importance of cargo receptors. We have just a few minor points that should be addressed:

Response: We very much thank the reviewer for the in-depth review of our study and for highlighting the conceptual advance it brings to the protein trafficking field.

1. In Fig. 3B, the changes in flow cytometry with siRNA treatment are modest, as the authors pointed out. It would be helpful if the authors can provide associated quantification of the data (e.g., number of cells in each replica and median/CV of the population distribution) to help the reader with the visualisation.

Response: We have added the relevant information (Figure 3B, S3A-B). We agree that the previous visualization was not optimal. Therefore, we divided the histograms in a main figure and supplemental controls. Furthermore, we provide all the requested metrics in the supplements and have added a quantification of the independent replicates.

2. As the total abundance of TLR4 is not affected by down regulation of TMED7, it might be interesting to see where the excess TLR4 localizes -- will this cause extra stress on the cells?

Response: The point raised an interesting question and we tried to address the impact of RHBDL4-mediated TMED7 processing on sub-cellular localization of TLR4. Unfortunately, all our efforts to detect TLR4 by immunofluorescence microscopy failed. We have tried to detect endogenous TLR4 in various cell lines with different antibodies. Moreover, we tested eight different commercially available antibodies (sc-13593, sc-13591, sc-52962 and sc-293072 by Santa Cruz, 48-2300 by Thermo Scientific, ab22048 and ab13867 by Abcam and 11811-1-AP by Proteintech) in the commercially available HEK-BLUE hTLR4 cells (Invivogen) that respond to TLR4 ligands due to TLR4 over-expression. As shown below, we were not able to detect any significant signal even for ectopically expressed TLR4 that was

not present as background staining in the parental HEK-Blue Null2 cells that do not express TLR4. Over all, the lack of a specific TLR4 antibody suitable for immunofluorescence analysis reflects what we have learned from contact with key colleagues in the TLR field.

Figure: Commercially available antibodies failed to detect ectopically expressed TLR4 in stable HEK-Blue hTLR4 cells. (A) Fluorescence micrograph of PFA-/NP40 fixed HEK-Blue hTLR4 cells using the Proteintech antibody 19811-1-AP does not show any specific signal that is not detected in HEK-Blue Null2 control cells. Co-staining with the ER marker BAP31 and the plasma membrane marker ZO-1. Scale bar, 20 μ M. (B) HEK-Blue hTLR4 cells respond to LPS treatment (0.1 μ g/ml for 24h) with secretion of an NF κ B-responsive secreted alkaline phosphatase reporter while the parental HEK-Blue Null2 cells do not respond as they lack TLR4 expression, means \pm SEM, n=2. Alkaline phosphatase activity in the supernatant of cells was assessed by measuring color change in Quanti-Blue solution (Invivogen) at OD_{655nm}.

To test whether knockdown of RHBDL4, TMED7 or RHBDL4/TMED7 causes additional stress in THP-1 cells, we have performed XBP1 Splicing assays that did not indicate elevated stress levels (Figure S3D).

3. The specificity of RHBDL4-mediated degradation of the p24 paralogs is remarkable. Does the specificity correlate with sequence similarities among the p24 proteins? It would be interesting to have a brief alignment of TMED sequences in the domain recognized by RHBDL4 (similar to that mapped by TMED9 domain swapping by the authors) to address this point in the discussion section.

Response: We have added a sequence alignment and a Two Sample Logo analysis (Figure S7A-B) clearly demonstrating sequence differences between cleaved TMED proteins and inert homologues, and commented on this in the discussion section.

4. Some minor typos surrounding Figure S1: Figure S1B doesn't have all the p24s, and the panel doesn't match the figure legend with regard to labelling; line 146, should refer to S1C; line 151 should refer to S1D.

Response: Mistakes have been corrected. Thanks for pointing them out to us.

Reviewer #2

In this study, the authors demonstrate that RHBDL4 is a new negative regulator of TLR4 by targeting and degrading TMED7, a member of the p24 family of COPII adaptor proteins, which counteracts the transport of TLR4 to the cell surface. The study primarily focused on an in vitro setting using human HEK293 cells, THP1 cells, and murine bone marrow-derived macrophages. However, the findings could be strengthened if the authors also included a pre-clinical model of sepsis instead of endotoxemia, as this would be a more relevant and clinically meaningful model.

Response: Although we agree that a preclinical model of sepsis (for example the Cecal ligation and puncture (CLP) assay) would be informative, unfortunately, our current animal license does not provide us with the relevant ethical approval to perform these experiments. Moreover, the CLP model has important limitations, as it is not satisfactory for the development of novel therapy strategies to treat septic patients which are most at risk of organ failure and death.

Finally, we would like to highlight that survival upon LPS injection is a widely used and reliable alternative that recapitulates many of characteristics of human sepsis in a highly controlled and standardized model. Over all, we believe that the data currently provided, including also new data on an in vitro Tuberculosis infection experiment (Figure 4E-F and S4), support our cellular study and model.

Additionally, the lack of an LD50 survival model is a limitation of the study.

Response: We appreciate the critique but would like to stress that the primary scope of our study was a cellular one; the animal studies are appended to corroborate what is largely a cellular and mechanistic study. As we observed significant differences in RHBDL4-associated survival and inflammatory responses in our LD100 LPS assay, we argue that a full assessment of LD50 survival, while desirable, is beyond the scope of our study. Moreover, extending the study would require a significant increase in the number of animals required for experimental procedures that are classified as "severe" according to EU Directive 2010/63. Hence, we feel that a smaller cohort of animals is justified within the context of the present paper and we restricted the analysis to a lethal LPS dose. We argue that this does not limit the validity of our claims. Finally, given our new data (Figure 5), which

intriguingly connect RHBDL4 polymorphisms with Kawasaki disease, we agree with the reviewer that future studies, outside of the scope of the present manuscript are warranted to further dissect the pathophysiological roles of RHBDL4 specifically within the context of Kawasaki-disease-associated inflammation and its sequelae.

The authors did not mention the impact of this pathway utilizing other ligands of TLR4, which could be important to consider.

Response: We thank the reviewer for this useful suggestion which we have addressed in full. We have now included THP-1 experiments with the synthetic TLR4 ligand CRX-527 and observed results consistent with our LPS-based studies (Figure S3E). This further bolsters the premise that RHBDL4 regulates the TLR4 axis.

To better understand the compartmentalization and trafficking of TLR4, the authors could have provided confocal microscopy data.

Response: Due to limitations of available antibodies, unfortunately this point also raised by reviewer #1 could not be addressed (see above).

It would also have been informative to assess TLR4-associated adaptor molecules and the transcription factor NF- κ B or AP-1 when establishing RHBDL4 as a negative regulator of TLR4.

Response: We now show that RHBDL4 knockdown increased NF- κ B activation by performing a luciferase assay on THP1-Lucia NF- κ B cell line (Figure S3C).

REVIEWER COMMENTS

Reviewer #1 (Remarks to the Author):

The changes made in the revised version, including new data on a disease-relevant mutation, improve the manuscript. I have no further comments and fully support publication of this interesting study.

Reviewer #2 (Remarks to the Author):

The authors have partially addressed my previous comments. They may also consider to address my subsequent critiques in the next round of revision. In the Discussion section, it would be valuable for the authors to disclose why they did not employ the preclinical model of sepsis.

Addressing this limitation would enhance the study's relevance to clinical applications. For better clarity in the results section, it would be beneficial to incorporate subtitles corresponding to each of the Figures as presented in their respective figure legend sections. This inclusion can significantly improve the organization and comprehension of the data presentation. As the study revolves around the RHBDL4 and TMED7 interaction, suggesting a mechanistic correlation, it is essential to expand the methodology beyond immunoprecipitation. Incorporating additional techniques such as BIAcore to quantitatively determine the interaction, specifically in terms of dissociation constant value (KD), would strengthen the study's depth. However, should technical limitations prevent the use of BIAcore, acknowledging this as a potential limitation in their study would be prudent. These revisions would enhance the overall clarity, depth, and applicability of the study, ensuring a more comprehensive understanding of the findings and their implications.

Reviewer #1:

The changes made in the revised version, including new data on a disease-relevant mutation, improve the manuscript. I have no further comments and fully support publication of this interesting study.

Response: We very much thank the reviewer for the positive feedback to our study.

Reviewer #2:

The authors have partially addressed my previous comments. They may also consider to address my subsequent critiques in the next round of revision. In the Discussion section, it would be valuable for the authors to disclose why they did not employ the preclinical model of sepsis. Addressing this limitation would enhance the study's relevance to clinical applications.

Response: We very much thank the reviewer for the constructive critique that helped us improve our manuscript. We now also followed the additional suggestion and mention in the discussion section that “that a preclinical model of sepsis was out of scope of the current study and will be needed to fully resolve the complexity of the underlying mechanism”.

For better clarity in the results section, it would be beneficial to incorporate subtitles corresponding to each of the Figures as presented in their respective figure legend sections. This inclusion can significantly improve the organization and comprehension of the data presentation.

Response: We have inserted subtitles accordingly.

As the study revolves around the RHBDL4 and TMED7 interaction, suggesting a mechanistic correlation, it is essential to expand the methodology beyond immunoprecipitation. Incorporating additional techniques such as BIAcore to quantitatively determine the interaction, specifically in terms of dissociation constant value (K_D), would strengthen the study's depth. However, should technical limitations prevent the use of BIAcore, acknowledging this as a potential limitation in their study would be prudent. These revisions would enhance the overall clarity, depth, and applicability of the study, ensuring a more comprehensive understanding of the findings and their implications.

Response: We agree with the reviewer that an understanding of the dissociation constant (K_D) of a mammalian Rhomboid protease for its natural substrate would be of interest to the rhomboid enzymology field (i.e., an audience distinct from the one that our MS addresses). However, we note that determining the dissociation constant (K_D) of an intramembrane protease and its transmembrane substrate is far from trivial and is indeed currently technically impossible. Hence, BIAcore is not feasible.

*So far, the experimental determination of an intramembrane protease K_D has only been achieved for the recombinant and purified *E. coli* rhomboid GlpG (which does not require additional protein cofactors) and only with a model substrate (Dickey et al. Cell 2013). Unfortunately, despite several attempts, no-one in the field has yet succeeded in establishing any comparable cell-free assay for mammalian RHBDL4. There are moreover other complications beyond the assay described above for a bacterial rhomboid. RHBDL4 assembles with additional transmembrane cofactors of the ER-associated degradation pathway such as erlins and E3 ubiquitin ligases (Bock et al. Cell Reports 2022) and hence potentially relies on these auxiliary components for its function. Its activity is also influenced by lipid composition (Paschkowsky et al. JBC 2018). Reconstituting such a system would be extremely demanding and require a timeframe of several years rather than weeks or months (if indeed there was a precedent for it being technically feasible). Finally, drawing any firm*

conclusions about the physiological situation likely would be difficult. Therefore, in our opinion, the most physiological and reliable way of studying the RHBDL4-substrate interactions currently is using cell-based assays. Accordingly, we provide several independent lines of evidence that TMED7 is a natural substrate of RHBDL4, which implies direct interaction:

- TMED7 was identified as a novel RHBDL4 substrate in an objective proteomic screen.
- Ablation of RHBDL4 leads to an increase in endogenous TMED7 steady-state levels (across different cell types and model organisms), an effect that can be rescued by RHBDL4 wt but not by the catalytic SA mutant.
- Co-expression of RHBDL4 with TMED7 leads to a distinct cleavage fragment, an activity that is strictly dependent on the rhomboid active site serine.
- Ubiquitinated TMED7 is stabilized by co-expression of the RHBDL4-SA mutant, which in turn enables co-purification with the catalytically dead enzyme in the above-mentioned co-IP-experiment. Of note, we have performed the IP in both directions and observe a robust interaction of the RHBDL4 substrate trapping mutant and TMED7.
- We have identified the scissile bond within the luminal domain of TMED7 and show that replacing the cleavage site region renders it a non-RHBDL4 substrate.

We note that the reviewer acknowledges that such BIAcore experiments between RHBDL4 and TMED7 may not currently be technically feasible and therefore address in the discussion section this limitation as follows: "Although the exact molecular mechanism of how RHBDL4 selects its substrates remains to be determined (i.e., the binding constant), our results from the cell-based assay indicate that it triggers the downregulation of a specific set of TMED/p24 proteins."

REVIEWERS' COMMENTS

Reviewer #2 (Remarks to the Author):

Authors have addressed my comments. I do not have further comments.